# The Effect of Two Bed Bath Practices in Cost and Vital Signs of Critically Ill Patients

**DOI:** 10.3390/ijerph18020816

**Published:** 2021-01-19

**Authors:** Chia-Hui Tai, Tsung-Cheng Hsieh, Ru-Ping Lee

**Affiliations:** 1Department of Nursing, Hualien Tzu Chi Hospital, Buddhist Tzu Chi Medical Foundation, Hualien 970473, Taiwan; chiahua@tzuchi.com.tw; 2Institute of Medical Sciences, Tzu Chi University, Hualien 970374, Taiwan; tchsieh@gms.tcu.edu.tw

**Keywords:** bed bath, cost, disposable wipes, soap and water, critical illness

## Abstract

By promoting personal hygiene and improving comfort, bed baths can decrease the risk of infection and help maintain skin integrity in critically ill patients. Current bed-bathing practices commonly involve the use of either soap and water (SAW) or disposable wipes (DWs). Previous research has shown both bed-bathing methods are equally effective in removing dirt, oil, and microorganisms. This experimental study compared the cost, staff satisfaction, and effects of two bed-bathing practices on critically ill patients’ vital signs. We randomly assigned 138 participants into 2 groups: an experimental group that received bed baths using DWs and a control group that received bed baths using SAW. We compared the bath duration, cost, vital sign trends, and nursing staff satisfaction between the two groups. We used the chi-square test and *t*-test for the statistical analysis, and we expressed the quantitative data as mean and standard deviation. Our results showed the bed baths using DWs had a shorter duration and lower cost than those using SAW. There were no significant differences in the vital sign trends between the two groups. The nursing staff preferred to use DWs over SAW. This study can help clinical nursing staff decide which method to use when assisting patients with bed baths.

## 1. Introduction

Healthcare-associated infections develop after patients receive treatment for various diseases and increase the length of hospitalization, mortality rate, and hospitals’ costs. Healthcare-associated infections include infections of the urinary tract and bloodstream as well as pneumonia [1,2,3]. Pathogens that cause healthcare-associated infections can enter a patient’s body through an invasive catheter [4]. Bed baths help maintain skin integrity and remove dirt and microbes present on the skin’s surface, decreasing the risk of infection and increasing patient comfort [5].

Patients admitted to intensive care units (ICUs) are unable to bathe because of their physical condition as well as their need for indwelling catheters and equipment to monitor vital signs. Therefore, the nursing staff must physically clean these patients using bed baths [6,7], which are generally performed using soap and water (SAW) [8]. A previous study reported 71% of nurses use the traditional bed-bathing method of SAW [8]. However, 62.2%–98.0% of basins used in giving SAW bed baths are contaminated with multidrug-resistant pathogens [9,10]. As a result, SAW is increasingly being replaced with disposable wipes (DWs) for giving bed baths [11,12,13].

Although bed baths can reduce the amount of dirt and microorganisms on patients’ skin [14], the bed-bathing process may create stress or anxiety for the patients. According to Lope, bed baths significantly increased the anxiety level and blood pressure in some patients with myocardial infarction [14]. Because patients in ICUs cannot bathe themselves [6,7], nursing staff must use bed baths to clean the patients in a way that does not negatively affect their vital signs.

Previous studies have found that using DWs can reduce the duration and cost of bed baths [6,7,15,16] as compared with SAW. Bed baths using DWs and SAW are equally efficient in the removal of microorganisms [13,16,17]; however, few studies have compared how the two bed-bathing practices affect vital signs [17]. We have observed in clinical practice that patients given bed baths with SAW experience agitation and physiological changes, such as increased blood pressure, increased heart rate, and chills. Using DWs for bed baths has been regarded as too expensive, and nurses have expressed concerns about the clinical effects of the two bathing methods. This study aimed to compare the duration, cost, effects on patients’ vital signs, and staff satisfaction of bed baths using DWs vs. SAW.

## 2. Materials and Methods

### 2.1. Study Design and Participants

We used an experimental design to compare the cost and effects of the two bed-bathing practices on critically ill patients’ vital signs. The experimental group received a bed bath using single-package DWs (3M; Cavilon, USA), and the control group received a SAW bed bath. We used a moderate effectiveness estimate with the G-power 3.1.9.2 software to estimate the sample size. Setting the power at 0.8 and α at 0.05, we estimated the sample size at 128. Because we projected 10% of the sample could be lost to follow-up, we increased the sample size to 142 and recruited 71 patients for each group.

We employed convenience sampling to enroll critically ill patients hospitalized in the medical ICU (MICU) at the Tzu-Chi Medical Center from June 2016 to March 2017. We included patients who met the following criteria: (1) admission to the MICU and ≥20 years old; (2) absence of skin diseases, such as scabies and dermatophytosis; (3) presence of at least one indwelling catheter such as a central venous catheter, urinary catheter, or endotracheal tube; and (4) informed consent form signed by patients or their family members. We randomly assigned participants by selecting a slip from a sealed envelope that corresponded to either the experimental or control group. The patients in each group were bathed using identical steps from head to toe and front to back: head and neck → trunk → four limbs → perineum → back. After being trained on bed-bathing, the ICU nurses and assistant nurses worked together to administer the bed baths. All routine care measures were identical between the two groups except the bed-bathing method.

### 2.2. Data Collection

We created structured data collection forms that included questions about the patients’ general characteristics, physiological markers, and vital signs; bath duration; cost; and staff satisfaction. To assess face validity for the data collection forms, we employed experts from four relevant domains, including one wound care nurse, one nurse manager, one nursing lecturer, and two clinical nurses. The patients’ general characteristics included their sex, age, medical history, diagnosis, catheter status, and other categorical variables. Physiological markers included the patients’ relevant clinical values upon admission to the ICU, such as vital signs, Acute Physiology and Chronic Health Evaluation (APACHE) II score, body mass index (BMI), and laboratory data, which were checked or filled in by the nursing staff. The bath duration was the length of time taken for the bed bath from start to finish. The total cost included the quantity of consumables, laundry, and nursing labor used per participant. We assessed staff satisfaction by administering a five-question survey to the nursing staff one week after the study concluded.

### 2.3. Study Procedure

The ethics committee of the hospital reviewed and approved the study’s protocol (IRB104-86-A). The principal investigator was responsible for explaining the study’s aim, duration, bed-bathing procedure, inclusion criteria, and process to the hospital’s staff and administration. We explained the study’s aim and methods to the patients or their family members during enrollment. After obtaining signed informed consent, we randomized the participants into an experimental group (bed baths using DWs) or a control group (bed baths using SAW). During the study, the research team examined the medical records and recorded the patients’ general information and physiological markers, the bath duration, and the quantity of consumables, laundry, and nursing labor used.

The patients were given bed baths once every 2 days, and the bath temperature was set at 40 °C for groups. We analyzed one bath given to each participant. An investigator observed the entire bath process and started a timer when the nurse used the first wipe (DWs group) or washcloth (SAW group) to begin cleaning the patient’s face. The timer was stopped after the nurse finished dressing the patient and changing the bedsheets. The amount of time needed to prepare the bath materials and clean up afterward was added to the total duration of each bed bath. If more than one employee was required, the investigator used separate timers for each employee and then added their time together. The quantity of consumables used for each bath was recorded. In addition, the investigators measured each patient’s body surface temperature, heart rate, systolic blood pressure, and oxygen saturation before and after bathing using the MICU’s SOLAR 8000 M physiological monitor. The body surface temperature was obtained behind the right ear, which about 3 °C lower than the core body temperature.

### 2.4. Statistical Analysis

We used the SPSS 19.0 statistical software (IBM, Armonk, NY, USA) for data analysis. We set the significance level *α*-value at 0.05 and used two-tailed tests. For the descriptive statistics, we expressed the categorical variables as frequency and percentages and the continuous variables as mean (M) and standard deviation. For the inferential statistics, we compared the differences in the general characteristics, physiological markers, bath duration, vital signs, and cost between the two groups. We used the chi-square test to compare the categorical variables and the *t*-test to compare the continuous variables between the two groups. We used the generalized estimating equation (GEE) to test the differences in vital signs between the two groups at different points in time.

## 3. Results

### 3.1. General Characteristics of the Participants

Our study ran from 13 June 2016, to 12 March 2017. During this time, 148 potentially eligible patients met the inclusion criteria. Six participants declined to participate, and four participants were excluded because they transferred out of the MICU within 24 h. Thus, we enrolled a total of 138 participants. Figure 1 shows the enrollment process. Participants were aged 24–96 years (M = 67.63, SD = 15.3). Seventy-eight participants (56.5%) were male and sixty (43.5%) were female. We used the chi-square test to compare the sex, chronic disease history, admission department, age, disease severity, and indwelling catheter status between the two groups and found no significant difference (*p* > 0.05). Table 1 summarizes the general characteristics of the participants.

### 3.2. Bath Duration, Cost, and Vital Signs

Table 2 shows the bath duration and costs for each group. The mean bath duration for all participants was 29.1 min (SD = 9.9). The shortest duration of any bath was 11.5 min, and the longest duration was 62.2 min. Two nurses administered each bed bath. The mean bath duration for the DWs group was 23.8 min (SD = 7.5) and for the SAW group was 34.4 min (SD = 9.2), which was a statistically significant difference (*p* < 0.01).

The cost of consumables for the DWs group was significantly higher than that for the SAW group (*p* < 0.01). However, the laundry and nursing labor costs for the DWs group were significantly lower than those for the SAW group (*p* < 0.01). Therefore, the overall cost of bed baths for the DWs group was significantly lower than that for the SAW group (*p* = 0.02).

Figure 2 shows the trends in body surface temperature, heart rate, systolic blood pressure, and oxygen saturation between the two groups from before the bed bath started until 30 min after it ended. A total of eight were recorded during this time period. There were no significant differences in the participants’ temperature (*p* = 0.07), heart rate (*p* = 0.37), systolic blood pressure (*p* = 0.42), and oxygen saturation (*p* = 0.20) between the two groups using the GEE method.

### 3.3. Staff Satisfaction

To assess staff satisfaction with the two bed-bathing practices, we administered a 5-question survey to 33 nurses and 2 assistant nurses 1 week after the study’s completion. The nurses’ mean age was 29.8 years (SD = 7.1), and the mean length of time worked in the MICU was 3.5 years (SD = 1.3). Table 3 shows the nursing staff’s satisfaction with the bed-bathing practices: 91.4% found DWs more convenient, 64.7% preferred DWs, and 94.3% reported spending less time on bed baths given with DWs. However, 71.4% of the staff thought SAW cleans patients better than DWs do.

## 4. Discussion

Bed baths are used to physically clean dirt and microorganisms from the skin of critically ill patients, thereby decreasing the risk of infection [5,6]. It is important for bed baths to be performed efficiently; therefore, we compared the bath duration, cost, vital sign trends, and staff satisfaction between bed baths using DWs and SAW.

We found the bath duration in the DWs group was significantly shorter at 23.8 min (SD = 7.5) compared with the SAW group at 34.4 min (SD = 9.2). In addition, the total cost of using DWs for bed baths was significantly lower than the total cost of using SAW. Our results were consistent with those of previous studies [6,7,15,16], possibly because the DWs group in our study had lower laundry and nursing labor costs. Therefore, we believe it is financially feasible to perform bed baths using DWs in medical intensive care units.

Bed baths may be a source of stress for patients [13,18,19]. In this study, 73.9% of the participants had endotracheal tubes and could not complete the patient satisfaction survey because of limitations. To assess patient stress, we measured the changes in patients’ vital signs before, during, and after bathing, including physiological indicators, body temperature, heart rate, systolic blood pressure, and oxygen saturation. Although there were no significant differences between the two groups, Figure 2 shows the control (SAW) group’s changes in vital signs during the bathing process were more wide-ranging than those in the experimental (DWs) group. These changes may have reflected significant discomfort in the patients.

Surveys administered after the study’s completion revealed the nursing staff believed bed baths using DWs were more convenient and took less time than SAW bed baths. Conversely, most nurses felt SAW cleans patients better than DWs do. These differences in staff satisfaction may be associated with personal preference and familiarity with a certain bath method. Coyer et al. [8] reported some nurses believe using soap and water for bed baths produces cleaner results than using other methods. However, bed-bathing methods may be associated with institutional policies, available resources, and patient preferences [20]. We recommend that institutions formulate technical operational procedures for bed baths and conduct continuous education and training to improve employees’ understanding of the importance of bed baths [8,20].

This study showed that although using DWs to give bed baths can increase the cost of consumables, it can also reduce the bath duration and lower the total cost. Additionally, nurses expressed greater satisfaction with using DWs to administer bed baths.

This study was limited in that it included only critically ill patients from a MICU and analyzed the results of only one bath per patient, possibly limiting the generalizability of the results. Future studies should examine different bath methods for a variety of patients and include multiple baths for each patient.

## 5. Conclusions

This study showed using DWs for bed baths can reduce the duration and total cost and improve the nursing staff’s satisfaction. Although vital sign trends in the DWs group were more stable than in the SAW group, there were no significant differences between the two groups. Based on these results, DWs may be a preferable method of giving bed baths in critical care settings.

## Figures and Tables

**Figure 1 ijerph-18-00816-f001:**
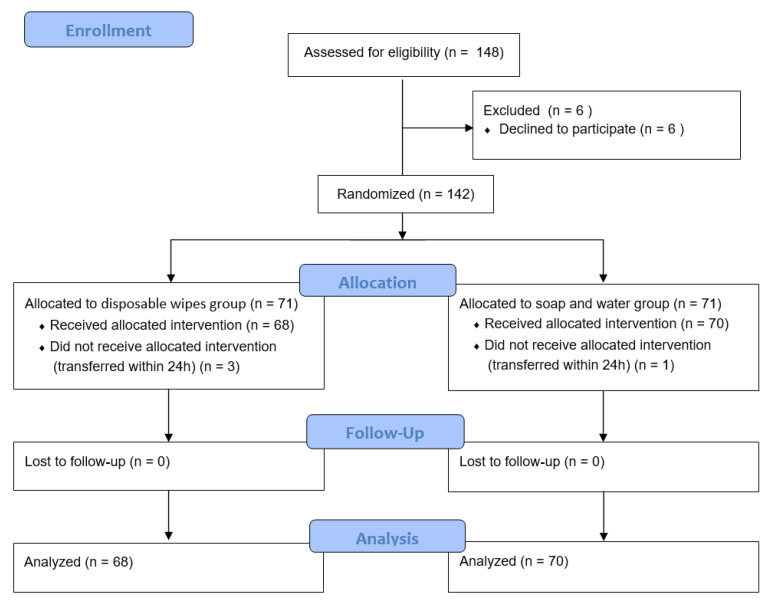
Enrollment process.

**Figure 2 ijerph-18-00816-f002:**
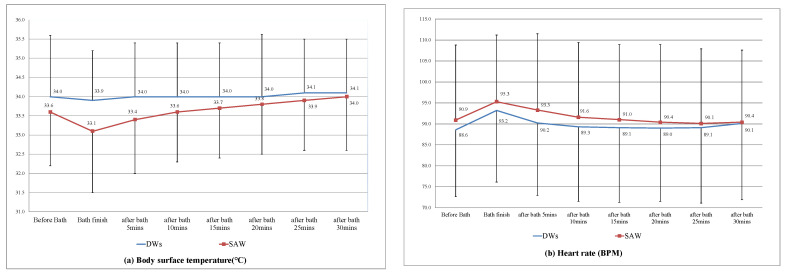
Vital sign trends for the two groups: (**a**) body surface temperature, (**b**) heart rate, (**c**) systolic blood pressure, and (**d**) oxygen saturation.

**Table 1 ijerph-18-00816-t001:** Comparison of the participants’ general characteristics.

Variable	Total Participants (*N* = 138)	DWs Bath (*n* = 68)	SAW Bath (*n* = 70)	*p*-Value
	*n* (%)	*n* (%)	*n* (%)	
**Age [*M* (SD)]**	67.6 (15.3)	65.4 (15.8)	69.6 (14.6)	0.11
**Sex**				0.88
Female	60 (43.5)	30 (44.1)	30 (42.9)	
Male	78 (56.5)	38 (55.9)	40 (57.1)	
**Diabetes History**				0.25
Yes	45 (32.6)	19 (27.9)	26 (37.1)	
No	93 (67.4)	49 (72.1)	44 (62.9)	
**Cardiovascular Disease History**				0.17
Yes	89 (64.5)	40 (58.8)	49 (70.0)	
No	49 (35.5)	28 (41.2)	21 (30.0)	
**Respiratory Disease History**				0.34
Yes	29 (21.0)	12 (17.6)	17 (24.3)	
No	109 (79.0)	56 (82.4)	53 (75.7)	
**Catheter Retention Status**				
Endotracheal tube	102 (73.9)	50 (73.5)	52 (74.3)	0.92
Central venous catheter	88 (63.8)	44 (64.7)	44 (62.9)	0.82
Urinary catheter	124 (89.9)	61 (89.7)	63 (90.0)	0.95
	***M* (SD)**	***M* (SD)**	***M* (SD)**	***p*-Value**
APACHE II	23.8 (7.0)	23.5 (8.0)	24.0 (6.0)	0.68
BMI	24.5 (5.9)	23.8 (5.4)	25.2 (6.4)	0.18
WBC (10^3^/uL)	13.0 (7.5)	13.1 (7.7)	13.0 (7.4)	0.97
Hb (g/dL)	10.6 (2.6)	10.6 (3.0)	10.7 (2.2)	0.79
PT (sec)	15.1 (11.4)	15.3 (13.5)	14.9 (9.1)	0.87
Na (mmol/L)	139.1 (7.6)	140.0 (8.8)	138.3 (6.2)	0.18
K (mmol/L)	4.2 (1.1)	4.0 (1.1)	4.3 (1.1)	0.19
BUN (mg/dL)	46.2 (39.9)	42.2 (41.1)	50.2 (38.6)	0.24
Creatinine (mg/dL)	2.4 (2.7)	2.0 (1.9)	2.8 (3.3)	0.11
Albumin (g/dL)	3.1 (0.6)	3.1 (0.5)	3.1 (0.7)	0.69
CRP (mg/dL)	9.9 (9.2)	9.4 (8.7)	10.3 (9.6)	0.61
Lactate (mg/dL)	3.8 (3.9)	4.1 (4.6)	3.4 (2.8)	0.31

APACHE II = Acute Physiology and Chronic Health Evaluation; BMI = body mass index; WBC = white blood cell; Hb = hemoglobin; PT = prothrombin time; BUN = blood urea nitrogen; CRP = C-reactive protein 3.2. Bath Duration, Cost, and Vital Signs.

**Table 2 ijerph-18-00816-t002:** Bath duration and costs of the two groups (*N* = 138).

	DWs Bath (*n* = 68)	SAW Bath (*n* = 70)	*p*-Value
Variable	*n* (%)	*n* (%)	
Duration of bath [min (SD)]	23.8 (7.5)	34.4 (9.2)	<0.01 *
Bath cost [NTD/bath (SD)]	237.9 (45.0)	255.3 (31.5)	0.01 *
Cost of bath consumables ^a^	124.9 (14.4)	66.7 (14.0)	<0.01 *
Clothing cost	33.7 (21.4)	74.1 (9.3)	<0.01 *
Cost of nursing time	79.2 (25.1)	114.5 (30.7)	<0.01 *

^a^ Bath consumable costs include DWs, soap, basin, and gloves * *p* < 0.05.

**Table 3 ijerph-18-00816-t003:** Staff satisfaction related to the bed-bathing methods (*n* = 35).

Item	DWs Bath *N* (%)	SAW Bath *N* (%)	No Difference *N* (%)	*p*-Value
Convenient use	32 (91.4)	0	3 (8.6)	<0.001 *
Lower bath duration	33 (94.3)	0	2 (5.7)	<0.001 *
More comfortable for patients	9 (25.8)	20 (57.1)	6 (17.1)	<0.001 *
Cleaner results	6 (17.2)	25 (71.4)	4 (11.4)	<0.001 *
Overall preference for bath method	22 (64.7)	7 (20.6)	5 (14.7)	<0.001 *

* *p* < 0.05.

## Data Availability

Data sharing is not applicable.

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
