# Peer review of "The Effect of Two Bed Bath Practices in Cost and Vital Signs of Critically Ill Patients"

_ijerph, 2021, doi:10.3390/ijerph18020816_

Round 1

Reviewer 1 Report

  1. there are multiple grammatical errors in the paper as well as some careless spelling mistakes such as vital sings. please do a thorough grammatical check to the whole manuscript.
  2. Research problem: the authors didn't explicitly explain the importance of measuring vital signs which is the major outcome of the study during bathing.
  3. subject recruitment: duration of subject recruitment was not mentioned; exclusion criteria is not mentioned. 
  4. I don't understand why the authors needed to conduct CVI for the registration form as the form is just for recording characteristics or clinical data of the subjects. it is not about a specific construct. It confused me.
  5. Questionnaire: it is unclear the number of items in each domain of the questionnaire. Why did they include staff satisfaction? it is not the research objective.
  6. how signed the consent? patient or family member?
  7. it is odd to suddenly that surface body temp was measured.
  8. it didn't tell when data collections were started. how many time  of data collections?
  9. the method sections are confusing. Please follow the CONSORT guideline to prepare the manuscript
  10. data analysis & discussion: no result between group and within group comparsion. I can't see the need of using GEE.

Author Response

Point 1: There are multiple grammatical errors in the paper as well as some careless spelling mistakes such as vital sings. please do a thorough grammatical check to the whole manuscript.

Response 1: The text has been modified in Line 16, 53, 61, 140, and 167.

Point 2: Research problem: the authors didn't explicitly explain the importance of measuring vital signs which is the major outcome of the study during bathing.

Response 2: This text has been supplemented in line 44 to 58.

Point 3: Subject recruitment: duration of subject recruitment was not mentioned; exclusion criteria is not mentioned. 

Response 3: This text has been supplemented in line 71 to 72.

Point 4: I don't understand why the authors needed to conduct CVI for the registration form as the form is just for recording characteristics or clinical data of the subjects. it is not about a specific construct. It confused me.

Response 4: Thank you for the reminder, we used face validity. Not content validity tests. The text has been modified in Line 88.

Point 5: Questionnaire: it is unclear the number of items in each domain of the questionnaire. Why did they include staff satisfaction? it is not the research objective.

Response 5:  The Questionnaire had five questions. This text had been supplemented in line 94.

Point 6: how signed the consent? patient or family member?

Response 6: Family member signed the consent. The text had been supplemented in Line 77.

Point 7: it is odd to suddenly that surface body temp was measured.

Response 7: We used the physiological monitor in the intensive care unit for continuous monitoring of vital signs. The body temperature measured is the body surface temperature behind the ear, which was about 3 degrees Celsius lower than the core body temperature. In order to provide continuous measurement, the body surface temperature was presented. The text had been supplemented in Line 114 to 115.

Point 8: it didn't tell when data collections were started. how many time of data collections?

Response 8: Study duration was June, 2016, to March, 2017. The text had been supplemented in Line 71 to 72.

Point 9: The method sections are confusing. Please follow the CONSORT guideline to prepare the manuscript

Response 1: The text had been supplemented in Line 138 to 139.

Point 10: data analysis & discussion: no result between group and within group comprasion. I can't see the need of using GEE.

Response 10: The text had been supplemented in Line 152 to 154.

Reviewer 2 Report

In times of pandemic, it is imperative for scientists to prioritize their research topics. The real, everyday social problems, despite their "simple" appearence, become the most important goals for our studies.  

Author Response

Point 1: Why the patients’ temperature was so low? Do you have the other metric system for this variable?

Response 1: We used the physiological monitor in the intensive care unit for continuous monitoring of vital signs. The body temperature measured is the body surface temperature behind the ear, which was about 3 degrees Celsius lower than the core body temperature. In order to provide continuous measurement, the body surface temperature was presented. The text had been supplemented in Line 114 to 115.

Reviewer 3 Report

The authors are suggested to specify whether the use of disposable towels from a registered trademark implied a conflict of interest.

Author Response

Point 1: The authors are suggested to specify whether the use of disposable towels from a registered trademark implied a conflict of interest.

Response 1.: Because there is only one of disposable wipe that meets the requirements in Taiwan.  Not to imply a conflict of interest.

Reviewer 4 Report

This manuscript provides us with a basis for decision-making for clinical nursing staff when assisting patients in bathing. In fact, the topic is very relevant for clinical care and the study design and the study procedure are very clear. The aim of the study is interesting. However, the study has a very small sample, and since it was conducted in a single bath, the authors must be careful when interpreting the results. In fact, some conclusions are very preliminary.

I would like to make some suggestions for revision:

Introduction

Line 47 - please provide references in the end of “…Few studies have compared the two bath practices in trends of vital sings”

Line 47- “ The objective of this study was to compare the cost and trends of vital sings in two bathing practices”  Why did you do that? Please revise the aim, taking into account the importance of this study for nurses' practice and for the patient. Why is it important to compare costs and trends in vital signs? The authors need to justify the importance of this study a little more.

Materials and methods  

Line 77 – “The CVI value …”

Line 80 - …” vital signs, APACHE II score, BMI,…”

Please spell out “CVI”, “APACHE II” and “BMI” for the readers to understand the names

Results

Line 115 – “The study period lasted from June 13, 2016, to March 12, 2017. During the study period, a total of 148 potentially eligible patients met the inclusion criteria. Six participants declined to participate, and 3 participants dropped out because they transfer out of the unit within 24 h. In total, 138 participants were included in the study. Figure 1 shows the enrollment process” .   This information is part of the study design. In the study design you wrote that 142 was the sample, but in the results you initially considered 148 participants. You must clarify the enrollment process in the results. The Figure 1 should be in the results section.

Line 144 –“… working in the ICU”- Please spell out “ICU” for the readers to understand the name

Table 1- “p value 00.18”- please write 0.18

Table 2 –“ Cost of nursing time” -How do you quantify the costs of nursing time? You need to clarify this aspect

Discussion

In general, the whole discussion is in need of improvement. You have to explain:

-Why were the total bathroom costs for the DWs group significantly lower than those for the SAW group?

-Why are the employees more satisfied with the DWs?

-Why do patients prefer DWs?

Line 174 –“ Bed baths may be a source of stress to patients [13, 17, 18]. In this study, 73.9% of the participants had endotracheal tubes and could not complete the patient satisfaction survey due to limitations. Physiological indicators, body temperature, heart rate, systolic blood pressure, and oxygen saturation, were used to observe the participants’ stress in this study. There was no significant difference between the two groups”. With a single bath you cannot conclude that, please rephrase this paragraph. Also rephrase the conclusion too.

Line 179- In this whole paragraph you repeat, too much, the word "However", please reformulate this paragraph taking into account other conjugations too.

Conclusions

Line 195-“The DWs group can reduce bath duration and total bath costs and employees have greater satisfaction following disposable bed bath”- You came to the conclusion, but what do you recommend, what has this information contributed to the practices of nurses? You need to improve the conclusion.

Author Response

Point 1:

Line 47 - please provide references in the end of “…Few studies have compared the two bath practices in trends of vital sings”

Line 47- “ The objective of this study was to compare the cost and trends of vital sings in two bathing practices”  Why did you do that? Please revise the aim, taking into account the importance of this study for nurses' practice and for the patient. Why is it important to compare costs and trends in vital signs? The authors need to justify the importance of this study a little more.

Response 1.: The text had been supplemented in Line 44 to 58.

Point 2:

Line 77 – “The CVI value …”

Line 80 - …” vital signs, APACHE II score, BMI,…”

Please spell out “CVI”, “APACHE II” and “BMI” for the readers to understand the names

Response 2.: The text has been modified in Line 91.

Point 3: Results

Line 115 – “The study period lasted from June 13, 2016, to March 12, 2017. During the study period, a total of 148 potentially eligible patients met the inclusion criteria. Six participants declined to participate, and 3 participants dropped out because they transfer out of the unit within 24 h. In total, 138 participants were included in the study. Figure 1 shows the enrollment process” .   This information is part of the study design. In the study design you wrote that 142 was the sample, but in the results you initially considered 148 participants. You must clarify the enrollment process in the results. The Figure 1 should be in the results section.

Line 144 –“… working in the ICU”- Please spell out “ICU” for the readers to understand the name

Table 1- “p value 00.18”- please write 0.18

Table 2 –“ Cost of nursing time” -How do you quantify the costs of nursing time? You need to clarify this aspect

Response 3.: The text has been modified in Line 55, 56, 130, 138, 139 and 159.

 Point 4: Discussion

In general, the whole discussion is in need of improvement. You have to explain:

-Why were the total bathroom costs for the DWs group significantly lower than those for the SAW group?

-Why are the employees more satisfied with the DWs?

-Why do patients prefer DWs?

Line 174 –“ Bed baths may be a source of stress to patients [13, 17, 18]. In this study, 73.9% of the participants had endotracheal tubes and could not complete the patient satisfaction survey due to limitations. Physiological indicators, body temperature, heart rate, systolic blood pressure, and oxygen saturation, were used to observe the participants’ stress in this study. There was no significant difference between the two groups”. With a single bath you cannot conclude that, please rephrase this paragraph. Also rephrase the conclusion too.

Line 179- In this whole paragraph you repeat, too much, the word "However", please reformulate this paragraph taking into account other conjugations too.

Response 4.: The text has been modified in Line 186, 187, 191, 192, 194 to 196, and 212 to 213.

Point 5: Conclusions

Line 195-“The DWs group can reduce bath duration and total bath costs and employees have greater satisfaction following disposable bed bath”- You came to the conclusion, but what do you recommend, what has this information contributed to the practices of nurses? You need to improve the conclusion.

Response 5.:The text has been modified in Line 216 to 219.

Round 2

Reviewer 4 Report

The authors clearly improved the manuscript, however, I strongly recommend that a native English speaker take a look at this paper, after an English revision it will be susceptible for publication

Author Response

Thank you for the reminder, our manuscript has edited for language, grammar, structure, and content, from the aspect of fluency and nativity.
